# Single-cell tumor-immune microenvironment of BRCA1/2 mutated high-grade serous ovarian cancer

I.-M. Launonen [1], N. Lyytikäinen[1], J. Casado [1], E. A. Anttila [1], A. Szabó[1], U.-M. Haltia[1,2], C. A. Jacobson [3], J. R. Lin[3], Z. Maliga[3], B. E. Howitt[4], K. C. Strickland[5], S. Santagata [3,6,9], K. Elias[7,8], A. D. D'Andrea [8], P. A. Konstantinopoulos [8], P. K. Sorger [3,9] & A. Färkkilä [1,2,3,8,10 ✉]

The majority of high-grade serous ovarian cancers (HGSCs) are deficient in homologous recombination (HR) DNA repair, most commonly due to mutations or hypermethylation of the *BRCA1/2* genes. We aimed to discover how *BRCA1/2* mutations shape the cellular phenotypes and spatial interactions of the tumor microenvironment. Using a highly multiplex immunofluorescence and image analysis we generate spatial proteomic data for 21 markers in 124,623 single cells from 112 tumor cores originating from 31 tumors with *BRCA1/2* mutation (*BRCA1/2*mut), and from 13 tumors without alterations in HR genes. We identify a phenotypically distinct tumor microenvironment in the *BRCA1/2*mut tumors with evidence of increased immunosurveillance. Importantly, we report a prognostic role of a proliferative tumor-cell subpopulation, which associates with enhanced spatial tumor-immune interactions by CD8+ and CD4 + T-cells in the *BRCA1/2*mut tumors. The single-cell spatial landscapes indicate distinct patterns of spatial immunosurveillance with the potential to improve immunotherapeutic strategies and patient stratification in HGSC.

[1] Research Program in Systems Oncology, University of Helsinki, Helsinki, Finland. [2] Department of Obstetrics and Gynecology, Helsinki University Hospital, Helsinki, Finland. [3] Laboratory of Systems Pharmacology, Harvard Medical School, Boston, MA, USA. [4] Department of Pathology, Stanford University School of Medicine, Stanford, CA, USA. [5] Department of Pathology, Duke University Medical Center, Durham, NC, USA. [6] Department of Pathology, Brigham and Women's Hospital, Boston, MA, USA. [7] Department of Obstetrics and Gynecology and Reproductive Biology, Brigham and Women's Hospital, Boston, MA, USA. [8] Dana-Farber Cancer Institute, Brigham and Women's Hospital, Harvard Medical School, Boston, MA, USA. [9] Ludwig Center for Cancer Research at Harvard, Harvard Medical School, Boston, MA, USA. [10] iCAN Digital Precision Cancer Medicine Flagship, Helsinki, Finland. ✉email: anniina.farkkila@helsinki.fi

Every year almost 240,000 new cases of ovarian cancers are diagnosed, leading to the death of over 150,000 women worldwide[1]. Out of all ovarian cancers, high-grade serous ovarian, peritoneal or fallopian-tube cancer (HGSC) is the most common subtype and the cause of 70–80% of all ovarian cancer-related deaths[2]. The 5-year overall survival of HGSC has remained poor, approximating less than 40%, with DNA-damaging therapy and cytoreductive surgery as the standard of care[2,3]. The development of new immunological therapies and patient stratification has been hampered by the lack of a detailed understanding of the tumor-immune microenvironment in HGSC.

Characteristically 20–22% of HGSC tumors are deficient in BRCA1 or BRCA2 genes, either via germline or somatic mutations, or hypermethylation[4]. BRCA1/2 mutations disrupt the high-fidelity homologous recombination (HR) DNA repair pathway, and consequently these cells are dependent on error-prone DNA repair mechanisms and sensitive to both DNA damaging agents and Poly-ADP Ribose Polymerase inhibitors[5]. In addition to the BRCA1/2 mutations, HR-deficiency can be caused by mutations or epigenetic modifications in other genes of the HR-pathway[5].

Notably, preliminary evidence suggests that HR-deficient tumors have a distinct tumor-immune microenvironment[6,7]. BRCA1/2mut tumors have been predicted to contain more neoantigens than tumors with no alterations in genes of the HR pathway (HRwt)[6], harbor an increased number of tumor-infiltrating lymphocytes[6], and have an elevated PD-L1 expression as compared to HR-proficient tumors[6,8]. Consistent with these findings, we recently reported enhanced responses to combination of PARP inhibitor plus PD-1 inhibitor in patients with HR-deficient tumors[9]. However, a comprehensive view of the tumor microenvironment (TME) and the interplay of tumor, immune and stromal cells of BRCA1/2 mutated tumors has not yet been described.

We performed highly multiplexed imaging utilizing 21 markers in 44 HGSCs and deep phenotypic profiling of 124,623 single cells to uncover the roles of the tumor- immune- and stromal cell subpopulations in the HGSC TME. We herein report phenotypically and spatially divergent microenvironments in the BRCA1/2mut tumors as compared to the HRwt tumors, with distinct cellular subpopulations and their spatial interactions contributing to clinical outcomes in BRCA1/2 mutated HGSC.

## Results

**Multiplexed imaging reveals TME composition at single-cell resolution.** We performed highly multiplexed imaging of 21 markers on a TMA with 112 cores (Supplementary Table 1) and quantified the data at single-cell resolution from 44 HGSCs (Fig. 1a). We annotated and categorized the single cells to tumor, immune and stroma, and subsequently to functional cell

subpopulations. The tumor microenvironment (TME) compositions, cellular diversity and functional states were associated to patient clinical outcomes. The clinical characteristics of the patients are shown in Table 1. As expected, the platinum-free interval (PFI), calculated as the time from last platinum-based chemotherapy to tumor relapse, was significantly longer in patients with BRCA1/2mut tumors as compared to patients with HRwt tumors in this patient cohort ($p = 0.02$, log-rank test, HR 0.30 (95% CI 0.13–0.70), Fig. S1c). Overall survival (OS) was not significantly different between the groups ($p = 0.3$, log-rank test, HR 0.38 (95% CI 0.11–1.26), Fig. S1d).

The markers used for annotating the cell types in the TME are shown in Fig. S1b. Prior to clustering with FlowSOM ($k = 20$), the data were sample-wise normalized using mean-centering. First, tumor cell clusters were separated, and the remaining cells were further clustered to detect and annotate clusters by their stromal and immune marker expression. The immune-annotated cells were first divided into lymphocytes and myeloid cells with FlowSOM ($k = 16$). Lymphocytes yielded four subtypes (CD8 + T-cells, CD4 + T-cells, CD4 + FOXP3 + T-regulatory cells and CD20 + B-cells) with PhenoGraph (knn = 60, markers CD8a, CD4, FOXP3 and CD20). Myeloid cells we assigned into six subtypes (IBA1 + CD163+, IBA1 + CD163 + CD11c +, IBA1 + CD11c +, IBA1 + and CD163+ macrophages, and CD11c + antigen-presenting cells) using FlowSOM ($k = 16$). Separately, tumor cells were clustered using k-means ($k = 25$, markers E-cadherin, Ki67, P21, PDL1, cCasp3, PSTAT1, and Vimentin) and annotated into seven metaclusters: epithelial cells, proliferating epithelial cells, epithelial-mesenchymal-transition (EMT) cells, proliferating EMT cells, mesenchymal cells, apoptotic cells, and functional epithelial cells. Stromal cell metacluster annotation with k-means (knn = 25, markers vimentin, CD31, Ki67, P21, PDL1, cCasp3, pSTAT1, and cell eccentricity – defined as the ratio of the distance between the focus points of an ellipse to its major axis length) produced nine metaclusters: high-proliferative stroma, proliferative stroma, non-proliferative stroma, high-vimentin, low-vimentin, low-eccentricity, functional stroma, high-p21 stroma, and endothelia.

Out of all 124,623 cells, 70% were classified as tumor cells, 15% immune cells, 12% stromal cells, and 3% remained unannotated (Images, cell segmentation masks, and single-cell data can be found at DOI: 10.7303/syn23747228). As expected, each tumor was characterized by a unique TME composition, with tumor cells being the dominant cell type in 40 out of 44 patients (Fig. 1b). UMAP shows that the tumor cells formed patient-specific clusters, whereas the immune, stromal and endothelial cells formed more common clusters between patients (Fig. 1c, d). To further characterize the tumor cells according to their functional states, we classified the 85,621 tumor cells into seven metaclusters using minimum spanning trees (Fig. S3a). The two most abundant metaclusteres were proliferating epithelial (29.9%) and epithelial (29.6%) cells, followed by EMT (15.3%) and proliferating EMT (11.4%) phenotypes (Fig. 1e, f), and each sample contained cells belonging to a minimum of five different tumor metaclusters (Fig. 1e).

We next classified the 19,697 immune cells into distinct immune-cell subtypes. Of the immune cells, IBA1 + CD163 + macrophages were among the most abundant (29.3%), followed by CD8 + T-cells (18.1%) (Fig. 1g). Each immune cell subtype showed consistent marker expression profiles (Fig. S1e) and visual clustering (Fig. 1h). Further, the immune cells followed the canonical trajectories in lineage analysis (Fig. S1f). Importantly, the proportions of CD3 + and CD8 + T-cells in the imaging-based single-cell data correlated with the scoring from conventional immunohistochemistry evaluated from fields selected for high CD3 + T-cell content, providing additional evidence on the

**Table 1 Summary of patient clinical characteristics.**

| Variable | Category | mean | median | range |
|---|---|---|---|---|
| Age at diagnosis (years) | All patients | 58.1 | 59.1 | 42.2–80.0 |
| | BRCA1/2 mutated | 55.2 | 53.7 | 42.2–75.6 |
| | HRwt | 65 | 64.3 | 49.1–80.0 |
| PFI (months) | All patients | 39.4 | 14.3 | 0.7–271.7 |
| | BRCA1/2 mutated | 49.5 | 24 | 6.2–271.7 |
| | HRwt | 15.3 | 6.9 | 0.7–110.4 |
| OS (months) | All patients | 66.4 | 44.6 | 5.8–276.7 |
| | BRCA1/2 mutated | 82.1 | 53.3 | 7.5–276.7 |
| | HRwt | 29 | 17.9 | 5.8–116.3 |

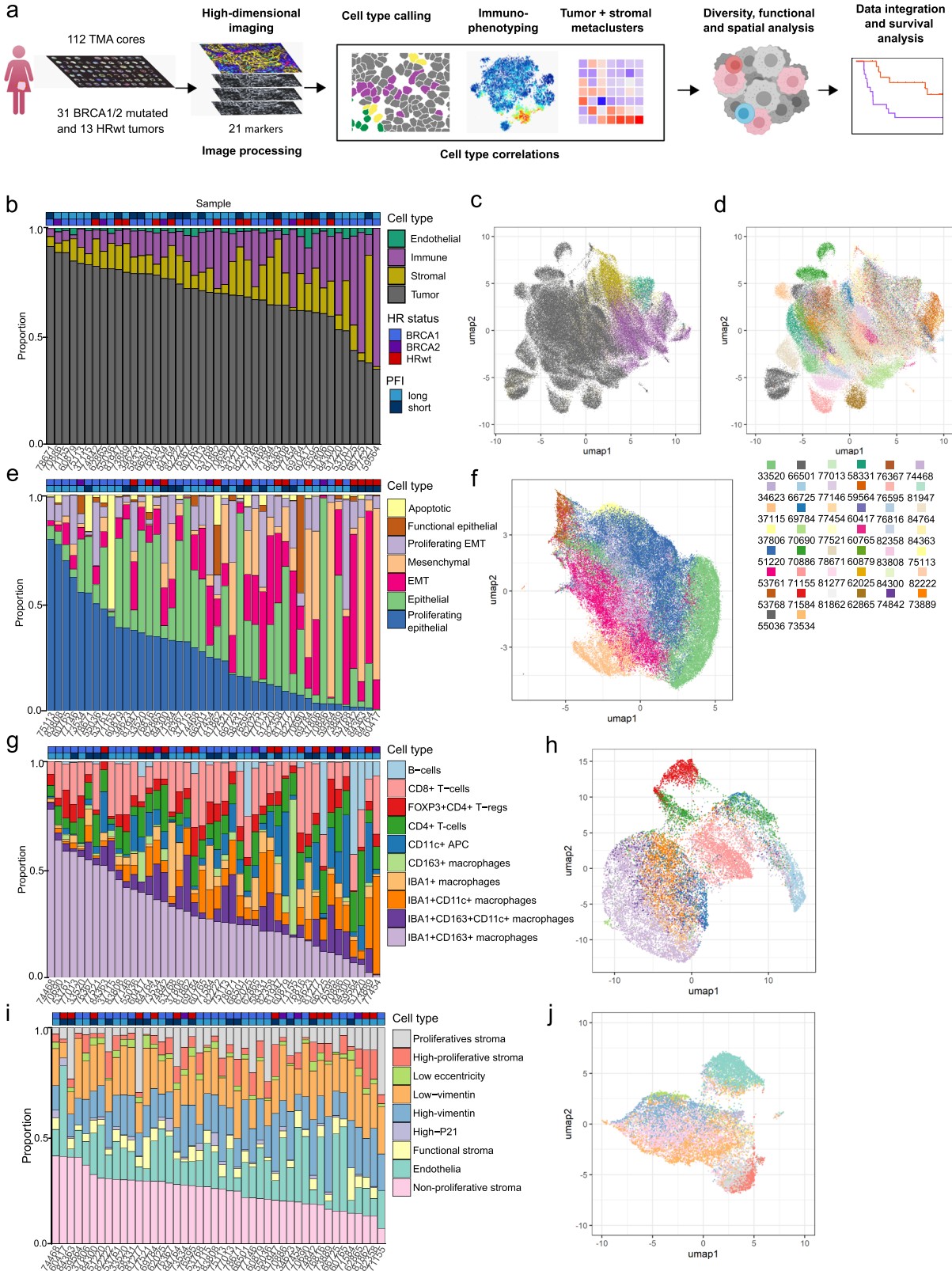

accuracy of the automated imaging-based cell type classification (Fig. S1h–j).

We next classified the 15,402 stromal cells into nine metaclusters using functional markers and morphological features (Fig. 1j, Fig. S1k). In contrast to tumor- and immune compositions, the stromal cells showed a heterogeneous composition without a dominant metacluster in any tumor (Fig. 1i).

However, a subset of the patients formed a cluster characterized by an enriched proportion of proliferative stromal cells (Fig. S1g). Furthermore, the BRCA1/2mut tumors had a higher overall ratio of tumor cells to stromal cells as compared to the HRwt tumors (Wilcoxon $p = 0.038$, $r = 0.29$, Fig. S1l).

Via iterative clustering approaches, we were able to separate the 124,623 single cells into phenotypic tumor metaclusters, immune

**Fig. 1 Multiplexed imaging reveals tumor microenvironment composition at single-cell resolution. a** A schematic representation of the workflow, from tumor tissue microarray (TMA) composition and high-dimensional imaging to cell type calling, population description, spatial and functional analysis, and clinical correlations. Using a TMA, a total of 112 cores from 31 patients with BRCA1/2mut tumors and 13 patients with tumors without any mutations in HR genes were imaged with 21 markers. After image processing, labels from cell type calling were used for description of immune subtypes as well as tumor and stromal metaclusters. These profiles were in turn used in diversity analysis. The spatial and functional states of cell types were analyzed and correlations with clinical information were evaluated. **b** Annotated barplot of the tumor, immune, stromal, and endothelial cell composition per patient, showing high tumor content across patients. The barplot is sorted by decreasing tumor proportion. Clinical annotations for barplots include HR status and PFI, with a PFI of < 12 months annotated as short and > 12 months long. **c** UMAP visualizing tumor, immune, stromal, and endothelial cells as distinct populations with color annotations corresponding to **b**, and **d** colored by their patient of origin (annotation panel below figure). The most abundant subtype were tumor cells, followed by stromal, immune, and endothelial cells. **e** Annotated barplot describing the tumor metacluster composition across different patients and clinical groups, showing heterogeneity across patients. The barplot is sorted by decreasing proliferating epithelial cell proportion. **f** UMAP visualizing the tumor cell metaclusters as distinct populations. **g** Annotated barplot describing the immune subtype composition across different patients and clinical groups, with a heterogeneous immune composition between patients. The barplot is sorted by decreasing IBA1 + CD163+ macrophage proportion. **h** UMAP visualizing the immune cells as distinct populations, with cells of lymphoid and myeloid lineages grouping to the opposite sides of the UMAP. **i** Annotated barplot describing the stromal metacluster composition across different patients and clinical groups, showing less heterogeneity across patients than tumor metaclusters and immune cell subtypes. The barplot is sorted by decreasing non-proliferative stroma content. **j** UMAP visualizing the stromal cell metaclusters as distinct populations. Source data are provided with this paper.

---

subtypes, and stromal metaclusters. Each tumor was characterized by a unique TME composition, with tumor cells forming the most abundant cell population in almost every tumor. In contrast to tumor metaclusters and immune subtypes, the stromal metaclusters showed no dominant metacluster in any tumor.

### Immune subtypes associate with immune diversity in HGSC HR-genotypes.

Profiling of the immune cell subpopulations revealed highly variable immune compositions across the patients. Using hierarchical clustering of the proportions of different immune cell subtypes out of all immune cells, we observed that a cluster of tumors harboring an IBA1 + CD163+ macrophage –dominant immune composition was almost exclusively BRCA1 mutated ($p = 0.045$, Fisher's exact test, Fig. 2a).

Consistently, the BRCA1/2mut tumors exhibited a significantly higher infiltration of IBA1 + , CD163 + and IBA1 + CD163 + macrophages as compared to the HRwt tumors (Wilcoxon $p = 0.027$, $r = 0.34$, Fig. 2b, Fig. S2a). Additionally, we noted a trend of increased infiltration of CD8 + T-cells in the BRCA1/2mut tumors when compared to the HRwt (Wilcoxon $p = 0.056$, $r = 0.29$, Fig. S2b). Deeper analysis of the immune compartment of the tumors revealed distinct myeloid cell profiles with the HRwt tumors exhibiting increased proportions of CD11c + antigen presenting cells (CD11c + APC, Wilcoxon $p = 0.0013$, $r = 0.47$), and IBA1 + CD163 + CD11c + macrophages (Wilcoxon $p = 0.00067$, $r = 0.49$) out of all immune cells as compared to the BRCA1/2mut tumors (Fig. 2c, d). These findings led us to explore the ratios of CD8 + T-cells to macrophages and antigen-presenting cells in the TME of the different HR-genotypes. Consistently, we observed an increased ratio of CD8 + T-cells to both IBA1 + CD163 + CD11c + macrophages (Wilcoxon $p = 0.0043$, $r = 0.46$, Fig. 2e) and to IBA1 + CD11c + macrophages (Wilcoxon $p = 0.0012$, $r = 0.43$, Fig. S2c) in the BRCA1/2mut tumors as compared to the HRwt tumors, indicative of enriched T-cell mediated immunosurveillance over CD11c + myeloid cell populations in the BRCA1/2mut tumors as compared to the HRwt tumors.

We next examined whether the proportions of the different immune cells associate with PFI or OS using a median cut-off for high or low proportions of the immune cell infiltration out of all cells. Consistent with previous findings[10,11], higher CD8 + T-cell infiltration was associated with longer PFI in all patients (Fig. 2g, $p = 0.015$, log-rank test, HR 0.27, 95% CI 0.11–0.63), but not with OS. However, this difference was not significant when the analysis was limited to only patients with BRCA1/2mut or HRwt tumors (Fig. S2d, e). Interestingly, a higher infiltration of CD4 + T-cells was associated with longer PFI, also separately in patients with

BRCA1/2mut tumors (Fig. 2h, $p = 0.0011$, log-rank test, HR 0.26, 95% CI 0.10–0.66, Fig. S2g). This finding was not significant in patients with HRwt tumors (Fig. S2f). Additionally, a high infiltration of FOXP3 + CD4 + T-regulatory cells was associated with an improved PFI in all patients as well as in patients with BRCA1/2mut tumors, and a high infiltration of IBA1 + CD163 + macrophages in all patients (Fig. S2h–m). However, in multivariate Cox regression models in patients with BRCA1/2mut tumors and in all patients pooled using the significant cell types from Kaplan-Meier analysis as variables, only CD4 + T-cells were significant and independently associated with PFI (BRCA1/2mut: $p = 0.043$, HR 0.24 (95% CI 0.06–0.95), All patients: $p = 0.02$, HR 0.34 (95% CI 0.14–0.85), Supplementary Table 2).

We next calculated a measure of intratumoral diversity for immune cell subtypes using Simpson's diversity index (SDI). Overall, the SDI for immune cells was higher in the HRwt tumors as compared to the BRCA1/2mut tumors (Wilcoxon $p = 0.0053$, $r = 0.41$, Fig. 2f), and the SDI did not correlate with the total immune cell infiltration (Fig. S2p). In the BRCA1/2mut tumors, the SDI for immune cells correlated positively with the proportions of IBA1 + CD163 + CD11c + (Spearman $p = 0.019$, $R = 0.42$, Fig. 2j) and IBA1 + CD11c + macrophages and CD11c + APCs (Spearman $p = 0.00002$, $R = 0.70$, Fig. S2n and Spearman $p = 0.00001$, $R = 0.71$, S2o), suggesting that a more diverse immune microenvironment associates with increased presence of cells with antigen presentation capacity. By contrast, SDI for immune cells correlated negatively with IBA1 + CD163 + macrophage proportions out of immune cells in both of the tumor HR-genotypes (BRCA1/2mut Spearman $p = 0.0014$, $R = -0.56$, HRwt: $p = 0.027$, $R = -0.62$ Fig. 2i). Immune diversity was however not associated with PFI in the two clinical groups (Fig. S2q–s).

Further investigation of the immune suppressive signals in the TME revealed an overall higher expression of PD-L1 in the macrophage and antigen presenting cell subtypes as compared to stromal (Wilcoxon FDR corrected $p < 2.2 \times 10^{-16}$) and tumor cells (Wilcoxon FDR corrected $p < 2.2 \times 10^{-16}$) (Fig. S2t). Importantly, the PD-L1 expression was significantly higher in the CD11c + APCs (Wilcoxon FDR corrected $p = 2.6 \times 10^{-9}$) and IBA + CD11c + macrophages (Wilcoxon FDR corrected $p < 2.2 \times 10^{-16}$) in the BRCA1/2mut, and in IBA1 + CD163 + CD11c + macrophages (Wilcoxon FDR corrected $p = 2.4 \times 10^{-5}$) in the HRwt tumors. Furthermore, the expression of PD1 was significantly higher in CD8 + T-cells (Wilcoxon FDR corrected $p = 0.00044$), CD4 + T-cells (Wilcoxon FDR corrected $p = 0.00044$), and FOXP3 + CD4 + T-regulatory cells (Wilcoxon FDR corrected $p < 2 \times 10^{-16}$) in the BRCA1/2mut as compared to HRwt tumors (Fig. S2u).

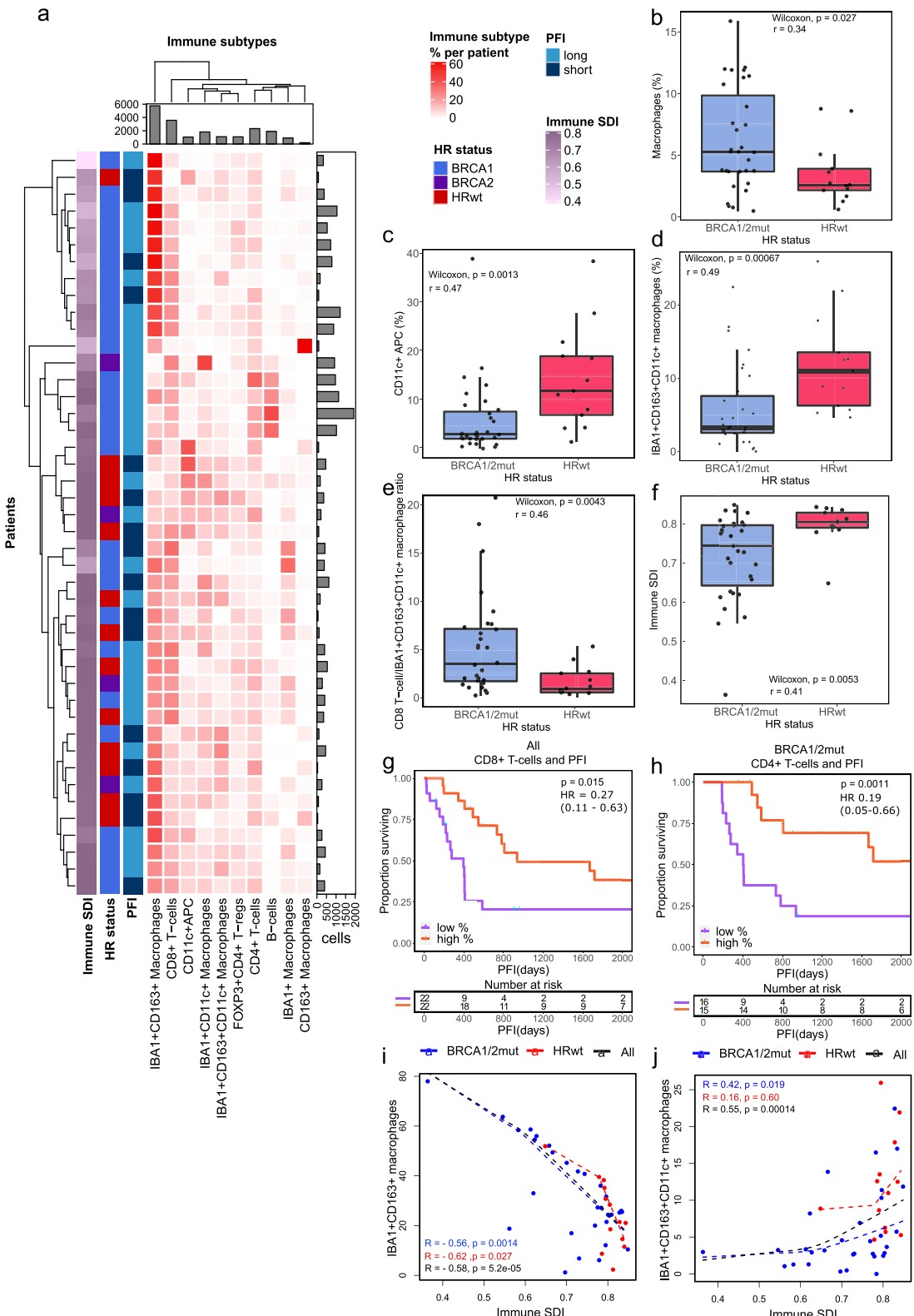

Functional marker expressions for the immune cell subtypes are presented in Supplementary Fig. 2v.

Altogether, the analysis of the immune microenvironment revealed marked differences in cell type compositions between the HR genotypes, with distinct myeloid cell compartments, associations to immune diversity and PD-1/PD-L1 expression profiles. The proportions of CD8 + T-cells associated with improved PFI

in all patients pooled, and CD4 + T-cells also separately in patients with *BRCA1/2*mut tumors.

**Functional tumor cells associate with distinct outcomes in the HR-genotypes**. We next examined the tumor single-cell phenotypes using tumor cell metaclusters annotated by their functional

**Fig. 2 Distinct immune microenvironments associate with immune diversity and clinical outcomes in HGSC. a** A hierarchical clustering heatmap of immune subtype proportions out of all immune cells, annotated with immune diversity (immune SDI), and clinical data including HR status and PFI. The barplot annotations for the columns and rows represent the total number of cells per each immune cell subtype and the number of the immune cells in total per patient, respectively. **b** Boxplot showing the combined proportion of IBA1 + , CD163 + and IBA1 + CD163 + macrophages as a proportion of all cells, stratified by HR status. **c** Boxplot showing the increased proportion of CD11c + antigen-presenting cells as well as **d** increased proportion of IBA1 + CD163 + CD11c + macrophages as a proportion of all immune cells in HRwt as compared to *BRCA1/2*mut tumors **e** Boxplot showing increased ratio of CD8 + T-cells to IBA1 + CD163 + CD11c + macrophages in *BRCA1/2*mut as compared to HRwt tumors. **f** Boxplot showing higher immune diversity in HRwt as compared to *BRCA1/2*mut tumors. The differences between the boxplot groups were calculated by two-tailed Wilcoxon rank-sum test. The black horizontal lines represent the sample medians, the boxes extend from first to third quartile and whiskers indicate the values at 1.5 times the interquartile range. Individual dots represent values per each tumor (n = 31 *BRCA1/2*mut, n = 13 HRwt). **g** Kaplan-Meier graphs for PFI show improved PFI for high CD8 + T-cell proportion in all tumors pooled as well as **h** for high CD4 + T-cell proportion in patients with *BRCA1/2*mut tumors. Proportions were calculated out of all cells, and the median value was used to distinguish high and low groups. Number of patients at risk is shown at the bottom of each Kaplan-Meier graph. The comparisons between the groups were performed using the log-rank test. **i** Scatter plots with lowess regression show negative correlation of immune diversity and the proportion of IBA1 + CD163 + macrophages in all HR groups and **j** a positive correlation of immune diversity and the proportion of IBA1 + CD163 + CD11c + macrophages in *BRCA1/2*mut tumors. Proportions were calculated out of immune cells and Spearman correlation coefficients and their p-values are shown (no FDR adjustment). HR groups include *BRCA1/2*mut (n = 31, blue), HRwt (n = 13, red) tumors and all tumors pooled (n = 44, black). Source data are provided with this paper.

marker expression. In hierarchical clustering of the proportions of tumor cell metaclusters out of all tumor cells, epithelial and proliferating epithelial metaclusters clustered together whereas the EMT and proliferating EMT formed a unique cluster, suggesting that either epithelial or EMT state is favored within each tumor (Fig. 3a). By visual inspection of a minimum spanning tree constructed from the original cell clusters, proliferating epithelial cell clusters and proliferating EMT cell clusters were separated from epithelial and EMT cell clusters into separate branches (Fig. 3b), underscoring the differential proliferative states between these groups (Fig. S3a). The functional marker expression of each metacluster is presented in Fig. 3d.

In Lineage trajectory analysis of the tumor cell functional states, the epithelial and EMT endpoints were projected on the opposite simplex corners (Fig. 3c). The epithelial cells were projected between functional and proliferating epithelial cells, while the EMT cells were projected between the mesenchymal and proliferating

EMT cells reflecting the continuous nature of the EMT process. The mesenchymal tumor cells showed a lineage towards the EMT cells as well as directly towards proliferating EMT cells. The apoptotic cells were projected between the proliferating epithelial and proliferating EMT cells. The lineage trajectory plots colored by Ki67 and vimentin are presented in Supplementary Fig. 3b, c.

Interestingly, the *BRCA1/2*mut tumors contained overall an increased proportion of proliferating epithelial cells out of all cells as compared to the HRwt tumors (Wilcoxon p = 0.0053, r = 0.38, Fig. 3e). Moreover, the HRwt tumors contained an increased proportion of EMT cells out of all cells as compared to the *BRCA1/2*mut tumors (Wilcoxon p = 0.024, r = 0.37, Fig. S3d), however the difference in the total EMT proportion out of all cells was not significant when taking into account also proliferating EMT cells (Fig. S3e). A subset of patients displayed a dominantly mesenchymal or epithelial phenotype as their most abundant tumor metacluster, and further the cluster characterized by high epithelial cell content associated with low tumor cell metacluster SDI (Wilcoxon p = 0.013, r = 0.43, Fig. 3a). The tumor cell metacluster proportions did not associate with clinical outcomes (data not shown). The tumor SDI was similar between the HR genotypes, and did not correlate with the proportion of tumor cells out of all cells (data not shown), or PFI (Fig. S3f–h).

Interestingly, the functional states of the tumor cells were differentially associated with PFI in the HR-genotypes (Fig. 3f-g). Notably, the expression of the proliferation marker Ki67 affected PFI in an opposite direction in the *BRCA1/2*mut tumors as compared to the HRwt tumors (Fig. 3f, g). In Kaplan Meier

analysis stratified by the highest 1/3rd of median Ki67 expression in the proliferating epithelial cells, a high expression of Ki67 was associated with a prolonged PFI (Fig. 3h, p = 0.029, log-rank, HR 0.12, 95% CI 0.02–0.88) in patients with *BRCA1/2*mut tumors. By contrast, we observed an opposite trend in patients with HRwt tumors, where a low expression of Ki67 in proliferating epithelial cells was associated with a better PFI (Fig. S3j, p = 0.013, log-rank, HR 0.10 95% CI 0.10–0.90).

Altogether, the tumor cells were characterized by an abundance of proliferating epithelial cells in the *BRCA1/2*mut tumors, and EMT cells in the HRwt tumors. Hierarchical clustering of the tumor metacluster proportions suggested that either epithelial or EMT lineage is favored within each tumor. Lineage trajectory analysis showed a continuum of the EMT process, and apoptotic cells stemming from proliferating epithelial and proliferating EMT cells. Importantly, we found an opposing prognostic role for Ki67 expression in proliferating epithelial cells in the HR genotypes, with a higher median Ki67 associating with an improved PFI in the *BRCA1/2*mut tumors.

**Cell subpopulations distinctively co-occur in the TME of the HR genotypes.** In hierarchical clustering of all cell-type proportions (Fig. 4a), the cell types clustered into two main groups, the first one characterized by tumor EMT and mesenchymal metaclusters, IBA1 + macrophages, and stromal metaclusters, and the second one characterized by other tumor metaclusters and immune cell subtypes, suggestive of a different patterns of immune activity in the tumor-rich and stromal-rich TMEs. Importantly, clustering analysis revealed differences in the overall cellular composition in the *BRCA1/2*mut and HRwt tumors (Fig. 4a). HRwt tumors were enriched in a cluster characterized by enhanced stromal presence from high-proliferative, high-P21, endothelial, non-proliferative, high-vimentin, functional, low-vimentin, low-eccentricity and proliferative stromal cells (p = 0.003, Fisher's exact test). Interestingly, as subset of the *BRCA1/2*mut tumors were characterized by high infiltration of CD8 + and CD4 + T-cells and B-cells.

To further investigate the interactions of the tumor metaclusters, immune subtypes and stromal metaclusters in the HR-genotypes, we plotted the correlations between the proportions of cell types separately for the *BRCA1/2*mut and the HRwt tumors (Fig. 4b). Overall, in the *BRCA1/2*mut tumors, the tumor metaclusters had more significant correlations with immune subtypes (n = 6) than with the stromal cell metaclusters (n = 0). By contrast, in the HRwt tumors, the stromal cell metaclusters had as many significant correlations with immune subtypes

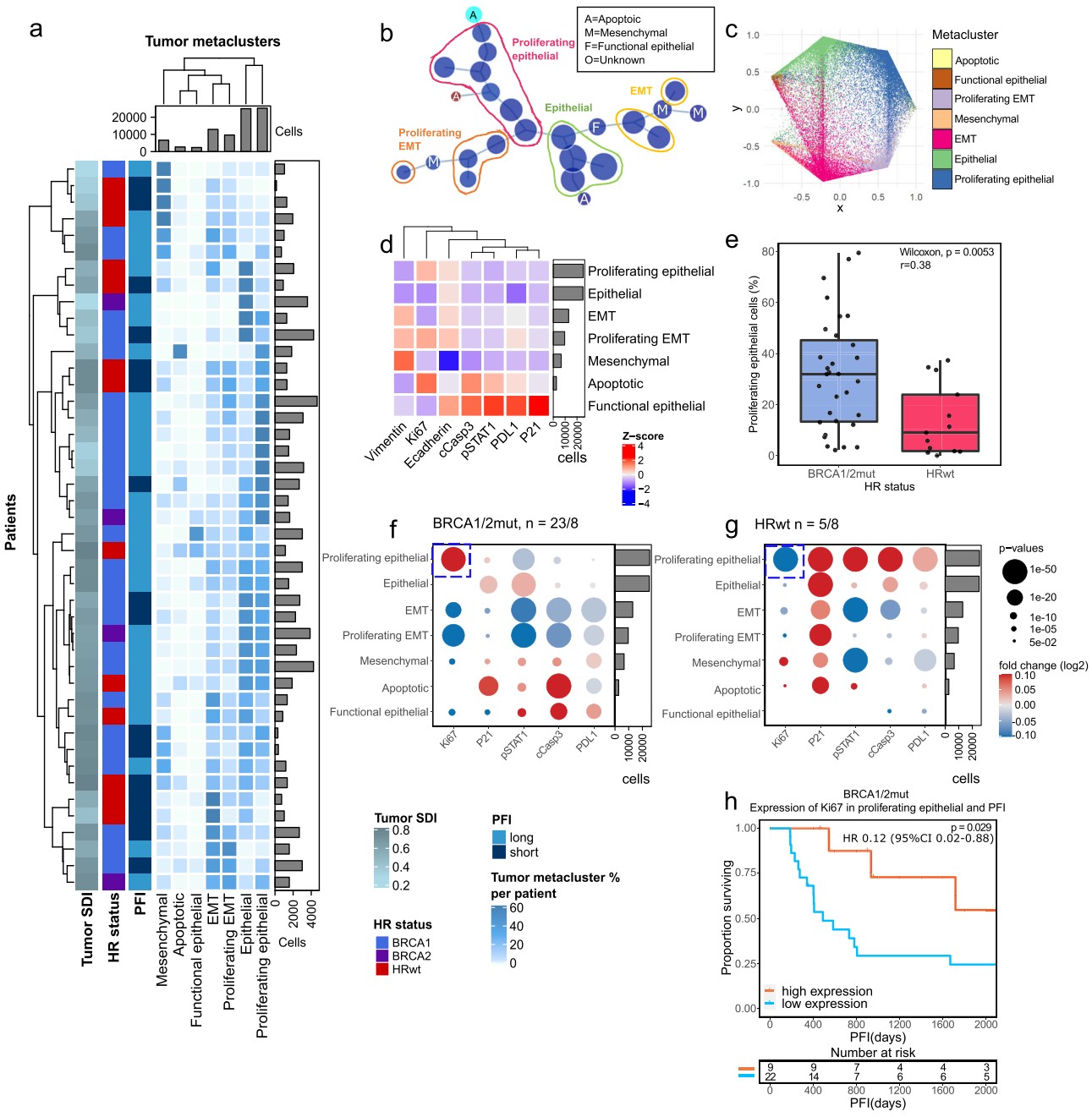

(n = 4) as with the tumor cell metaclusters (n = 4), suggesting distinct shaping of the immune microenvironment by the tumor cell subpopulations in the *BRCA1/2*mut tumors.

Overall, the proportions of CD8 + T-cells and CD4 + T-cells were highly correlated with each other in both of the HR-genotypes (*BRCA1/2*mut: Wilcoxon FDR corrected $p = 2.1 \times 10^{-5}$, HRwt: $p = 0.069$), and had similar correlations with other cell types (Fig. 4b). In the *BRCA1/2*mut tumors, CD8 + T-cells correlated positively with IBA1 + CD11c + macrophages (Wilcoxon FDR corrected $p = 0.0004$), CD4 + T-cells correlated positively with CD11c + APCs (Wilcoxon FDR corrected $p = 0.00022$), and both CD8 + T-cells and CD4 + T-cells correlated positively with B-cells (Wilcoxon FDR corrected $p = 2.5 \times 10^{-9}$, $p = 6.8 \times 10^{-5}$) suggestive of a TME with

coordinated immune activity. In the HRwt tumors, high-proliferative stroma correlated negatively with CD4 + T-cells (Wilcoxon FDR corrected $p = 0.07$), and the CD4 + T-cells correlated positively with IBA1 + CD163 + macrophages (Wilcoxon FDR corrected $p = 0.047$), suggesting that the stromal compartment may contribute to enhanced immunosuppression in the HRwt tumors.

To further explore the relationships between the cell types, we computed Principal Component Analysis (PCA)-plots separately for the *BRCA1/2*mut and the HRwt tumors (Fig. 4c, d). In both the *BRCA1/2*mut and the HRwt tumors, the first principal components were mainly characterized by either high immune infiltration or high stromal infiltration. In the *BRCA1/2*mut tumors, the second principal component was mainly

**Fig. 3 Tumor metacluster functional states associate with clinical outcomes in BRCA1/2mut tumors. a** A hierarchical clustering heatmap of tumor metacluster proportions out of all tumor cells, annotated with tumor diversity (SDI), HR status, and PFI. The barplot annotations for the columns and rows represent the total number of cells per each metacluster and the number of tumor cells in total per patient, respectively. **b** Minimum-spanning tree of the original tumor cell clusters with circles visualizing the assigned metaclusters based on median marker expression profiles. Smaller metaclusters are marked with letters. Annotations include the assigned metaclusters. **c** Lineage trajectory plot taking all tumor metaclusters as endpoints shows both separation and interconnections between epithelial and EMT phenotypes. **d** Heatmap of tumor metaclusters with expression levels (Z-score) of the markers in the metaclusters. Rows of the heatmap are sorted by metacluster abundance. **e** Boxplot showing increased proportion of proliferating epithelial cells as a proportion of all cells in the BRCA1/2mut ($n = 31$) as compared to HRwt ($n = 13$) tumors. The differences between the groups were calculated by two-sided Wilcoxon rank-sum test. Boxplots visualize the sample medians, first to third quartile and the values at 1.5 times the interquartile range. Individual dots represent values per each tumor. **f** Dot plot of fold changes (log2) of functional marker expression in tumor metaclusters in BRCA1/2mut tumors, between long ($n = 23$) and short ($n = 8$) PFI and **g** in HRwt tumors between long ($n = 5$) and short ($n = 8$) PFI. The color of the dots represents the fold change and their size the significance of the p-value. Fold changes with FDR corrected p-values < 0.05 are shown. Blue boxes highlight the differential Ki67 expression in proliferating epithelial cells in long and short PFI tumors, in the different HR-genotypes. **h** Kaplan-Meier graphs showing high Ki67 expression in proliferating epithelial cells associates with an improved PFI in patients with BRCA1/2mut tumors. Patients belonging to the top 1/3 percentile in all tumors in median Ki67 expression were annotated as high ($n = 9$) and the rest as low ($n = 22$). Number of patients at risk is shown at the bottom of the Kaplan-Meier graph. The comparison between the groups was performed using the log-rank test. Source data are provided with this paper.

characterized by proliferative epithelial and mesenchymal tumor metaclusters, with proliferating epithelial cells opposing the mesenchymal metacluster cells and the immune subtypes. In the HRwt tumors, the second principal component was characterized by stromal metaclusters and the mesenchymal tumor metacluster, which oppose the proinflammatory IBA1 + CD11c + macrophages. The similar direction of the loading vectors of immune cells in the PCA-plots, especially in the BRCA1/2mut tumors, suggested a coordinated immune system with simultaneous recruitment of multiple immune cell subtypes. Consistently, the PCA biplot indicated that patients with a long PFI generally harbored tumors with rich immune infiltration, whereas patients with a short PFI were characterized by tumors rich in stromal metaclusters (Fig. 4e, f). The third principal component revealed similar direction of EMT, proliferating EMT and high-proliferative stromal cell metaclusters opposing the epithelial cell metacluster in the BRCA1/2mut tumors (Fig. S4a, b). In HRwt tumors, the third principal component showed opposing direction of the epithelial cell metacluster and CD11c + APCs, FOXP3 + CD4 + T-regulatory cells as well as functional and low-eccentricity stroma (Fig. S4c and d), further supporting the stromal contribution on immunosuppression in the HRwt tumors.

The PCA-plots with all patients pooled further supported the distinct TMEs in the two HR-genotypes; the first two principal components display the HRwt tumors characterized by stromal metaclusters and the BRCA1/2mut tumors characterized by immune cell subtypes and proliferating epithelial cells (Fig. S4e and f). The third principal component, characterized by EMT and proliferating EMT metaclusters opposing CD163 + macrophages and endothelial cells, showed further separation of the HR genotypes (Fig. S4g, h).

Altogether, the combined analysis of all cell types showed differential TME composition and cell type co-occurrence profiles in the HR genotypes. CD8 + and CD4 + T-cells correlated positively with other immunostimulatory cells suggestive of coordinated recruitment of immune cells in the BRCA1/2mut tumors. By contrast, the HRwt tumors were characterized by an increased presence of stromal metaclusters, their negative association to CD4 + T-cells, and role in opposing immune cell infiltration in the PCA-plots.

**Enhanced immunosurveillance with spatial interactions in BRCA1/2mut tumors.** To uncover the physical interactions between the cell types, we explored the TME spatial organization by calculating the fraction of neighboring cell types for each single cell and computed the mean of these fractions per tumor to compare the cellular communities between the BRCA1/2mut and HRwt tumors. The neighborhood fractions were normalized by their cellular abundancies in the corresponding tumor cores. Interestingly, we observed marked differences in the spatial organization of the TME cell types in the BRCA1/2mut as compared to the HRwt tumors. The correlation matrix and summary graphics are shown in Fig. S5a, and Fig. 5a. We evidenced enhanced spatial interaction of the proliferating epithelial cells with CD8 + T (Wilcoxon FDR corrected $p = 0.036$), CD4 + T-cells (Wilcoxon FDR corrected $p = 0.053$) and FOXP3 + CD4 + T-regs (Wilcoxon FDR corrected $p = 0.070$) in the BRCA1/2mut tumors as compared to HRwt tumors. Also, endothelial cells (Wilcoxon FDR corrected $p = 0.024$) and functional stromal cells (Wilcoxon FDR corrected 0.036) showed enhanced spatial interactions with the proliferating epithelial cells in the BRCA1/2mut tumors (Fig. 5a). Visual inspection of representative images revealed CD8 + T-cells residing in the neighborhoods of proliferating epithelial cells in a BRCA1/2mut tumor, whereas the TME was more compartmentalized with fewer cellular interactions between the proliferating epithelial cells and CD8 + T-cells in a HRwt tumor (Fig. 5b).

In the BRCA1/2mut tumors, the CD4 + T-cells resided predominantly in cellular communities rich in proliferating epithelial cells (Wilcoxon FDR corrected $p = 0.065$) and other CD4 + T-cells (Wilcoxon FDR corrected $p = 0.049$). Further supporting the spatial immune-stimulatory TME of the BRCA1/2mut tumors, the immune-supppressive FOXP3 + CD4 + T-regs showed enhanced spatial interactions with non-proliferative stromal cells (Wilcoxon FDR corrected $p = 0.070$).

We found no significant differences in immune cells neighboring endothelial cells between the two HR genotypes. However, endothelial cells were in enhanced spatial proximity with IBA1 + CD163 + macrophages significantly more often as compared to other immune cell subpopulations (Fig. S5c).

Additionally, we observed distinct spatial interactions between the myeloid cells and other cells of the TME. In the BRCA1/2mut tumors, IBA1 + CD163 + macrophage communities were enriched in proliferating epithelial cells and epithelial cells (Wilcoxon FDR corrected $p = 0.036$, $p = 0.056$). By contrast, the CD163 + macrophages neighbored functional stromal compartments in the HRwt tumors (Wilcoxon FDR corrected $p = 0.0007$), suggesting that these macrophages reside in functional stromal niches more often in the HRwt than in the BRCA1/2mut tumors. Hierarchical clustering of significant bidirectional interactions further highlights the differential spatial TMEs in the HR genotypes (Fig. S5b). Altogether, the distinct spatial cellular communities suggest enhanced immunosurveillance of

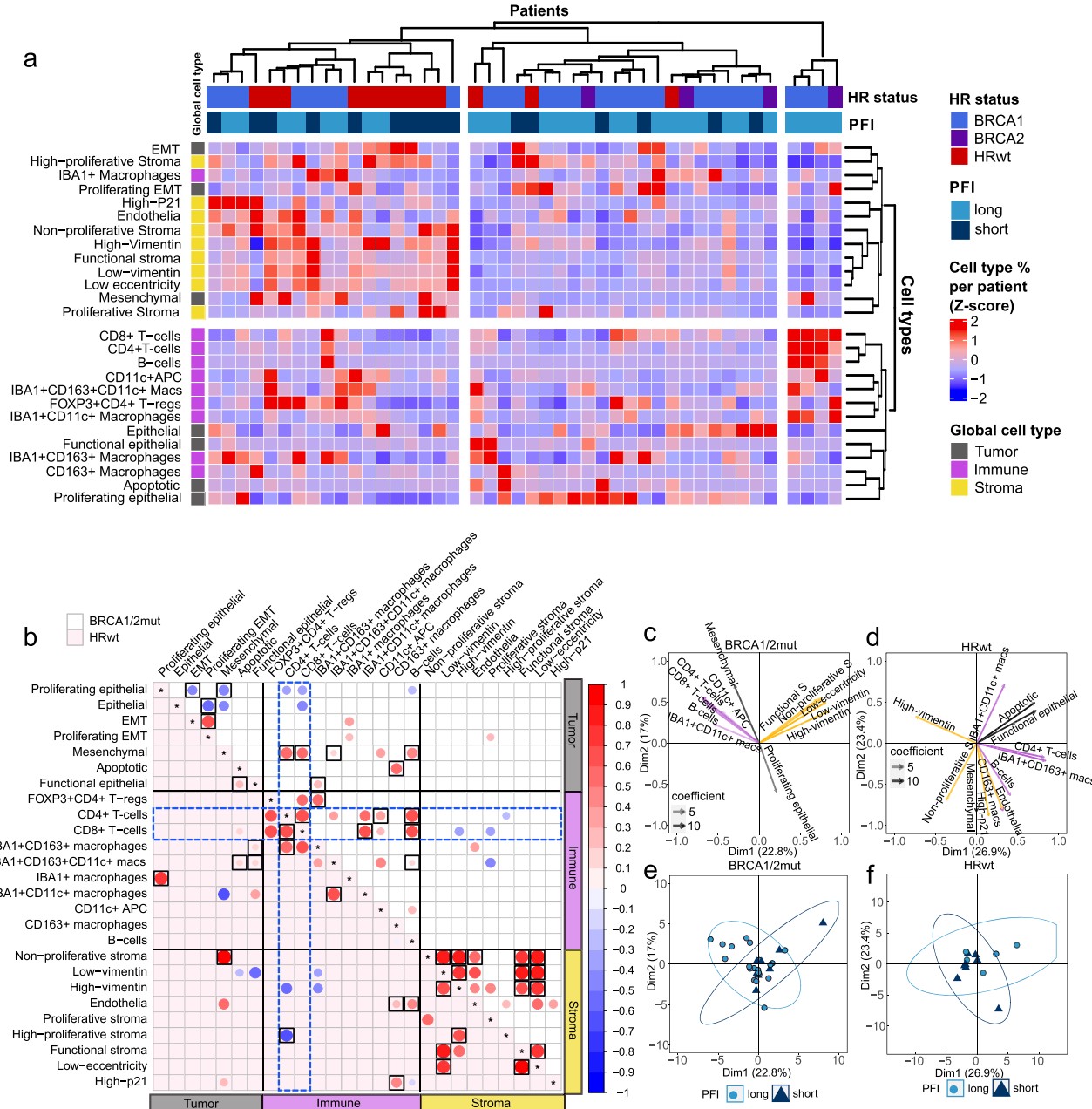

**Fig. 4 HR-genotypes show different patterns of cell type co-occurrence with CD8 + T-cells and CD4 + T-cells. a** A hierarchical clustering heatmap of all cell proportions shows enrichment of HRwt tumors in the cluster characterized by stromal metaclusters. Annotations include the HR status, PFI group, as well as the global cell type of the metaclusters. **b** A Spearman correlation dot plot between tumor metaclusters, immune subtypes and stromal metaclusters as proportions out of all cells, divided by HR status (*n* = 31 *BRCA1/2*mut, *n* = 13 HRwt). Dots in white background and light-red background represent correlations in *BRCA1/2*mut and HRwt tumors, respectively. Dots shaded red represent positive correlations while dots shaded blue represent negative correlations. Black lines separate the tumor, immune and stromal cell columns and rows. Blue dashed boxes highlight the interactions that CD4 + T-cells and CD8 + T-cells have with other cell types. Correlations with a p-value < 0.05 are shown and those passing FDR 0.1 are highlighted with black squares. **c** Principal component feature projections for BRCA1/2mut (*n* = 31) tumors and **d** for HRwt (*n* = 13) tumors. 12 cell types with the highest contributions to PC1 and PC2 are shown for each plot. S = stroma, macs = macrophages. The more intense the color of the cell type, the coefficient is. Colors represent global cell types (grey = tumor, yellow = stroma, purple = immune). **e** Plot of patients with BRCA1/2mut (*n* = 31) tumors and **f** plot of patients with HRwt (*n* = 13) tumors projected onto their first two principal components. Patients are annotated as having a long or short PFI and colored by light blue or dark blue, respectively. Ellipsoids show the 95% confidence intervals for the PFI groups. Source data are provided with the paper.

proliferating epithelial tumor cells by the CD4 + and CD8 + T-cells in the *BRCA1/2*mut tumors as well as enhanced spatial interactions of M2-macrophages with epithelial cells, and immune-suppressive M2-macrophage-stromal interactions in the HRwt tumors.

Finally, we investigated the potential prognostic roles of the spatial interactions of CD4 + T-cells and CD8 + T-cells with proliferating epithelial cells. We found that both the fractions of CD4 + T-cells and CD8 + T-cells in the neighborhood of proliferating epithelial cells were significantly associated with

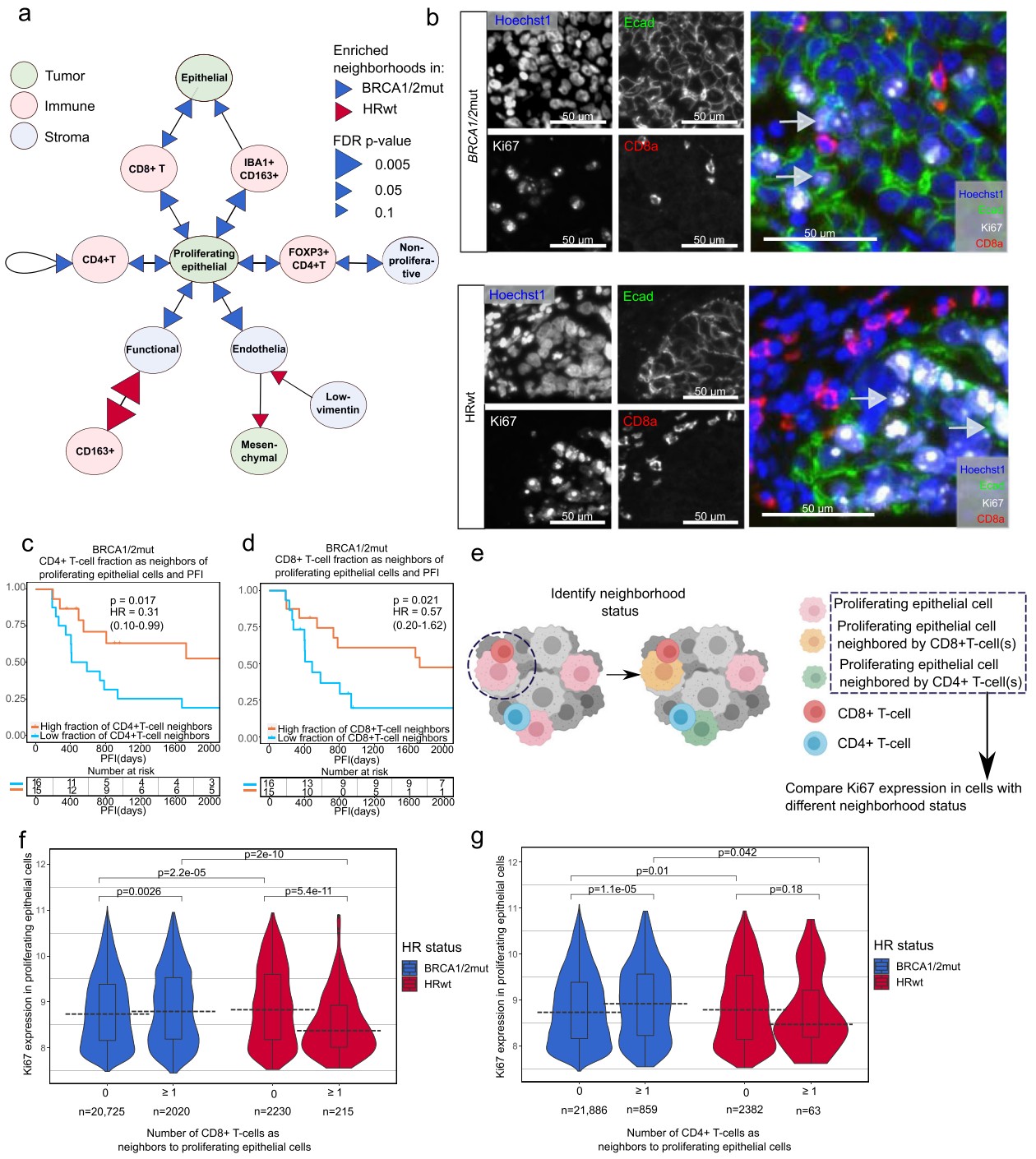

prolonged PFI in patients with *BRCA1/2*mut tumors (CD4 + T-cells: *p* = 0.017, log-rank, HR 0.31, 95% CI 0.10–0.99, CD8 + T-cells: *p* = 0.021, log-rank, HR 0.57, 95% CI 0.20–1.62, Fig. 5c, d). In a multivariate cox regression analysis using the CD4 + and CD8 + T-cell neighbor fractions of proliferating epithelial cells as well as the Ki67 status of proliferating epithelial cells, none of these were, however, independently significantly associated with PFI or OS (Supplementary Table 3). This suggests that, in addition to the Ki67 expression in epithelial tumor cells, the coordinated immune surveillance in the spatial cellular communities contributes to better prognosis in the *BRCA1/2*mut tumors.

**Immune cell spatial interactions with distinctively proliferating tumor cells.** The prognostic roles of CD8 + and CD4 + T-

cells, and their observed enhanced spatial interactions with proliferating epithelial cells in the *BRCA1/2*mut tumors prompted us to investigate whether Ki67 expression in proliferating epithelial tumor cells is associated with enhanced cell-to-cell spatial interactions. We compared the Ki67 expression in the proliferating epithelial tumor cells neighbored by CD8+ or CD4 + T-cells as compared to cells without these cells as neighbors (Fig. 5e). Interestingly, we observed a higher Ki67 expression in the proliferating epithelial cells with one or more CD8 + T-cells as neighbor as compared to cells without any CD8 + T-cells as neighbors in the *BRCA1/2*mut tumors (Wilcoxon *p* = 0.0026, *r* = 0.02), but not in the HRwt tumors (Fig. 5f, Fig. S5d). A similar phenomenon was observed in proliferating epithelial tumor cells with CD4 + T-cells as neighbors in patients with *BRCA1/2*mut

**Fig. 5 Proliferative state delineates spatial cell-cell interactions of the tumor-immune microenvironment in *BRCA1/2*mut and HRwt tumors. a** A schematic representation of cell-cell spatial interactions significantly enriched (FDR < 0.1) in *BRCA1/2*mut tumors ($n = 31$, blue arrows) and in HRwt tumors ($n = 13$, red arrows). The arrowheads point to the cell type surrounding the cell type denoted by the arrow base. The size of the arrows indicate the FDR-corrected p-value. The dotplot in Fig. S5a denotes the corresponding fold change values. **b** Representative multiplex immunofluorescence images showing highly proliferating epithelial cells co-localizing with CD8 + T-cells in *BRCA1/2*mut tumors, and highly proliferating epithelial cells residing further away from CD8 + T-cells in HRwt tumors as shown in **a**. Experiment was repeated for $n = 31$ *BRCA1/2*mut tumors and $n = 13$ HRwt tumors. **c** Kaplan-Meier graphs showing high fraction of CD4 + T-cells and **d** high fraction of CD8 + T-cells neighboring proliferating epithelial cells associate with an improved PFI in patients with *BRCA1/2*mut tumors. Median was used as a cut-off for high and low fractions. Number of patients at risk is shown at the bottom of each Kaplan-Meier graph. The comparisons between the groups were performed using the log-rank test. **e** A schematic illustration of functional state and spatial cell-cell interaction analysis. Proliferating epithelial cells were each characterized by their CD4 + T-cell and CD8 + T-cell spatial neighborhood status. Next, Ki67 marker expression was compared in proliferating epithelial cells with different cell-cell spatial neighborhoods in *BRCA1/2*mut and HRwt tumors separately. **f** Violin plots showing the probability densities of Ki67 expression in proliferating epithelial cells in the different CD8 + T-cell neighborhood and HR-groups. **g** Violin plots showing the probability densities of Ki67 expression in proliferating epithelial cells in the different CD4 + T-cell neighborhood and HR-groups. In the box plots inside violin plots the black horizontal lines represent the sample medians, the boxes extend from first to third quartile and whiskers indicate values at 1.5 times the interquartile range. Two-sided wilcoxon rank-sum test was used for pairwise comparisons. Source data are provided with the paper.

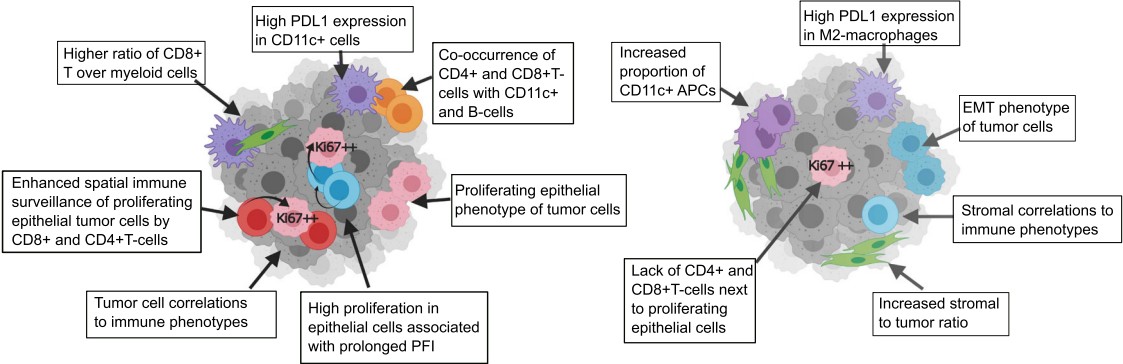

**Fig. 6 Graphical summary of the differences in the TME of *BRCA1/2*mut and HRwt tumors.** A schematic representation of the differences in cell type infiltration and their spatial interactions in the *BRCA1/2*mut and HRwt tumors, characterized by proliferating epithelial and EMT tumor cells, macrophages and antigen presenting cells, as well as the spatial interactions of CD8 + T-cells and CD4 + T-cells with differently Ki67 expressing proliferating epithelial cells and other immune cells. Curved arrows represent a spatial interaction.

(Wilcoxon $p = 1.1 \times 10^{-5}$, r = 0.03) but not with HRwt tumors (Fig. 5g, Fig. S5e). Collectively, these findings suggest that in the *BRCA1/2*mut tumors, high Ki67 expression in proliferating epithelial cells results in improved immunosurveillance, with enhanced cell to cell interaction with CD4 + T-cells and CD8 + T-cells, whereas in the HRwt tumors the high-proliferative tumor cells are not spatially interacting with CD4 + or CD8 + T-cells. These distinct spatial tumor-immune interactions provide potential insights into the mechanisms driving the differential prognostic impact of Ki67 expression in proliferative epithelial cells in tumors with different HR-genotypes.

## Discussion

Detailed information on the interplay of tumor, stromal and immune cells in the tumor microenvironment in HGSC has been lacking. Herein, we aimed to shed more light into how *BRCA1/2* mutations and single-cell tumor phenotypes shape the immune microenvironment and prognosis in HGSC. Using imaging-based single-cell data, we captured and annotated 124,623 cells of the tumor, stromal and immune cell populations in the TME of 44 HGSC patients. We discovered distinct immune microenvironments and divergent interactions between the immune subtypes and tumor and stromal metaclusters in tumors with *BRCA1/2*mut as compared to HRwt tumors. Interestingly, we found opposing prognostic roles of proliferative functional states of epithelial tumor cells in patients with *BRCA1/2*mut and HRwt tumors, which also associated with distinct cellular neighborhoods. The

graphical summary of the distinct TME phenotypes in the *BRCA1/2*mut and the HRwt tumors is presented in Fig. 6.

Single-cell immunophenotyping revealed divergent immune cell compartments in the HR genotypes. Consistent with previous observations from syngeneic mouse models[7], the *BRCA1/2*mut tumors contained an increased infiltration of IBA1 + and M2-type IBA1 + CD163 + and CD163 + macrophages, whereas the HRwt tumors had abundant CD11c + myeloid cells. The myeloid cells also showed abundant and distinct PD-L1 expression profiles between the HR genotypes, with the highest expression in the CD11c + cells in the *BRCA1/2*mut tumors and M2-type macrophages in the HRwt tumors. Overall, the infiltration of CD8 + and CD4 + T-cells correlated positively with the infiltration of various macrophages, as suggested also previously[12]. Further, consistent with previous reports[6,7,13], we observed a trend towards increased CD8 + T-cell infiltration in the *BRCA1/2*mut tumors, and a balance favoring CD8 + T-cell infiltration over infiltration of multiple myeloid cell types in the in the TME of the *BRCA1/2*mut tumors, suggesting enhanced immunosurveillance capacity via CD8 + T-cells. Importantly, the CD8 + T-cells were characterized by a higher PD-1 expression in the *BRCA1/2*mut tumors implicating potentially enhanced susceptibility to PD-1/ PD-L1 targeted immune checkpoint therapies. By contrast, the lower CD8 + T-cell infiltration patterns strengthen the notion that T-cell exclusion is more common in the HRwt tumors.

Investigation of the prognostic roles of the immunophenotypes revealed that a higher infiltration of both CD8 + T-cells and CD4 + T-cells were associated with prolonged responses to

platinum-based chemotherapy, consistent with previous findings[10,11]. Importantly, in multivariable analysis the CD4 + T-cell infiltration displayed an independent prognostic role, also in patients with *BRCA1/2*mut tumors. Interestingly, the CD4 + T-cells had more immune stimulatory correlations with CD11c + myeloid cells and B-cells in the *BRCA1/2*mut tumors. In addition, the proportions of CD11c + antigen presenting cells correlated with increased diversity of other immune cells, and the co-occurrence of tumor-infiltrating T-cells, B-cells, and antigen presenting cells suggested a more coordinated immune cell activity in the *BRCA1/2*mut tumors. Further, spatial analysis revealed enhanced self-interactions of the CD4 + T-cells in the *BRCA1/2*mut tumors as compared to the HRwt tumors. Altogether, these finding suggest a coordinated, immune-stimulatory role for CD4 + T-cells in the *BRCA1/2* mutated HGSCs.

Lack of single-cell resolution on large-enough clinical cohorts has hampered the in-depth investigation on the role of tumor cell subpopulations and their functional states on patient prognosis. Previous studies have reported increased proliferative states associated with *BRCA1* mutation in breast[14,15], and ovarian tumors[16]. However the reports on the potential prognostic role of high Ki67 expression in HGSC have remained conflicting[17–19]. In the present study, the proliferating epithelial cells formed the most abundant tumor cell subpopulation, which was further enriched in the *BRCA1/2*mut tumors. Interestingly, a high Ki67 expression in proliferating epithelial cells conveyed an opposing prognostic effect in the HR-genotypes; high expression of Ki67 in the proliferating epithelial cells associated with a longer PFI in patients with *BRCA1/2*mut tumors, but with a trend for shorter PFI in the HRwt tumors. This difference can be, at least in part, explained by the differential platinum sensitivity due to HR-deficiency: the highly proliferating epithelial cells are likely to be more sensitive to platinum-based chemotherapy in the *BRCA1/2*mut tumors as compared to the HRwt tumors. Thus, using single-cell resolution information in tumors stratified by the predicted HR capacity, we were able to resolve the previously conflicting conclusions, and revealed potentially clinically significant role of tumor cell functional proliferative state with the premise to promote the development of prognostic and predictive biomarkers for HGSC.

Our unique single-cell dataset enabled further characterization of the associations of the cellular functional states and their spatial interactions. Analysis of the cellular neighborhood frequencies revealed a higher proliferative state of the epithelial cells residing in CD4 + or CD8 + T-cell cellular communities in the *BRCA1/2*mut tumors. By contrast, the epithelial cells with a higher proliferative state were residing outside of the CD4 + or CD8 + T-cell neighborhoods in the HRwt tumors. The underlying mechanism behind the enhanced spatial immuno-surveillance is likely to involve increased interferon activation[20], and increased neoantigens[6] in the *BRCA1/2*mut tumors. In line with previous reports[12,21], the proportions of proliferating epithelial tumor cells correlated negatively with CD8 + and CD4 + T-cell infiltration, indicating that the immune surveillance of CD8 + and CD4 + T-cells towards the highly proliferating epithelial cells in the *BRCA1/2*mut tumors is regulated by spatial attraction and not just by their cellular abundance.

Consistently, the enhanced cellular communities of both CD4 + T-cells and CD8 + T-cells with the proliferating epithelial cells associated with a better PFI in patients with *BRCA1/2*mut tumors. Further, the prognostic role of the CD8 + T-cells was conveyed only by their spatial arrangement, and not by relative abundance in the *BRCA1/2*mut tumors. Importantly, the multivariate survival analyses confirmed that the prognostic role of the Ki67 expression in the proliferating epithelial cells is not independent of the spatial CD8 + and CD4 + T-cell communities,

implicating that coordinated and spatial interplay of the cell subpopulations is needed to convey prognostic relevance. Our single-cell spatial analyses of the TME thus illuminate the mechanisms on how both the tumor cell genotypes and immune infiltration contribute to prognosis in HGSC[22].

The role of the stromal cells in HGSC genotypes has also remained elusive[12,23]. We observed that the HRwt tumors contained an increased stromal to tumor ratio, and significant intercorrelations of the stromal metaclusters with the immune cell subpopulations, suggesting enhanced stromal contribution to the immunophenotypes and T-cell exclusion. Stromal cells promote the presence of physical and biochemical barriers which can further contribute to therapy resistance[12,24,25]. Moreover, we found that the HRwt tumors contained an increased proportion of tumor cells with an EMT phenotype, which also could contribute to chemoresistance[26–28]. Altogether, the stromal dominance, tumor cell EMT functional states, and decreased immune surveillance can contribute to the poor prognosis of the HRwt tumors.

We acknowledge certain limitations of the study. Our patient cohort consisted of a limited number of HRwt tumors as compared to the *BRCA1/2*mut tumors, rendering the conclusions made on the *BRCA1/2*mut tumors stronger. Further, most of the *BRCA1/2*mut tumors were *BRCA1* mutated ($n = 27$). Although the TME analysis using a tissue microarray minimizes experimental variation, it poses a potential bias on the conclusions regarding tumor heterogeneity as only a maximum of three 1 mm cores can be examined, and thus we had to refrain from performing permutation-based spatial analyses which require a larger dataset size, and focus on the communities based on normalized neighbor fractions for each cellular community. Further, the experimental design requires the application of a limited set of protein-based markers, thus leaving a small proportion of the cells (average 5% per tumor) unannotated. In addition, the pioneering nature of the work has hampered availability of similar cohorts or datasets, and thus validation in an external cohort, preferably using an unbiased method such as spatial single-cell RNAsequencing, would be needed to strengthen the conclusions of the study.

In conclusion, single-cell analysis highlights the functional and spatial differences in the TME landscapes of *BRCA1/2*-deficient and HRwt tumors, underscoring the need for differential treatment strategies for these clinical subgroups. Uncovering the effects of tumor genotypes on the TME, cellular phenotypes, and spatial communities will enable the development of more effective immunotherapeutic strategies and improved patient stratification in HGSC.

## Methods

**Cohort description.** The study was approved by the Mass General Brigham Institutional Review Board. Informed consent was waived due to the use of archival samples and anonymization of the material. The clinical cohort consisted of 27 patients with *BRCA1* mutated tumors (20 germline, five somatic, and one not determined), 4 patients with *BRCA2* mutated tumors (3 germline, 1 somatic), and 13 patients with tumors without any alterations in the HR-genes. HR-genes included *ATM, ATRX, BRCA1, BRCA2, BRIP1, CHEK2, FANCA, FANCC, FANCD2, FANCE, FANCF, FANCG, NBN, PTEN,* and *U2AF1*. The patients were annotated using targeted sequencing panel in a previous study[6]. Tumors harboring mutations in the nucleotide excision repair and mismatch repair pathways were excluded. In addition, immunohistochemistry for *BRCA1* was used to exclude *BRCA1* promoter hypermethylation in tumors without mutations in the HR genes[6]. The histology of all tumors was confirmed as HGSC by an experienced pathologist. The tumor material consisted of 112 tumor 1.0 mm cores on a single tissue-microarray (TMA); mostly triplicate cores per patient.

**Highly multiplexed imaging, and image processing.** The samples were stained sequentially with the validated antibodies (Supplementary Table 1) and scanned with RareCyte CyteFinder scanner following the tCycIF protocol[29]. Scanned image files were corrected using the BaSiC tool and stitched and registered using the

ASHLAR algorithm (https://github.com/labsyspharm/ashlar) to align image tiles and successive images of tiles from all cycles to one another. We detected and cropped the cores from the stitched multi-channel image with Image Processing Toolbox on MATLAB (version 2018b, The MathWorks, Inc., Natick, Massachusetts, United States). This process produced multi-channel *.tiff* files that were further converted for Ilastik using the Fiji plugin ilastik4ij-1.7.3. Probability maps for segmentation were produced using Ilastik, where all pixel features were used for class discrimination. We trained Ilastik with 10 cores on 3-class manually labeled data (inner nuclei, cell membrane, image background). We segmented the resulting probability maps (exported as unsigned 16-bit tif images with axis order tzyxc = [1, 1, 1400, 1400, 32]) and the simple segmentation results for the label 'nuclei'. The results were processed in Fiji to produce a final set of masks by expanding nuclei masks and removing objects with area < 30 pixels. We quantified log mean fluorescence intensity for each channel as well as default morphological features with histoCAT software. The mean number of cells per core was 1107, median 1030, and range 29–3282. Subsequently, the mean number of cells per sample was 2818, median 2524, and range 278–5933. The number of cells per patient and the number of TMA-cores per patient is presented in Supplementary Fig. 1A. We used three channels imaged with secondary-only antibodies (Rabbit 488, Rat 555, Mouse 647) with a background threshold of 97th percentile for quality control to remove unqualified cells. We removed lost cells based on radical changes in the DNA channel across cycles as previously described in Cycif Analysis Suite (https://github.com/yunguan-wang/cycif_analysis_suite). ROIs with artefacts (e.g., bubble, blobs, debris) were identified by visual inspection in the Open Microscopy Environment Omero, and were removed from further analysis. XY-plots of the marker expressions per single cell to were further visualized to confirm data quality.

**Cell phenotyping and spatial analysis**. We utilized our recently published tool CYTO[30] to cluster and iteratively assign cell type labels. The quantified cell data were clustered in CYTO, followed by annotation of the clusters based on their median marker expression profiles. The clusters were visualized with heatmaps and minimum spanning trees of marker expressions per cluster in CYTO, and with UMAP using marker expression per cell. The process consisted of five stages: (1) separation of tumor cells from immune and stromal cells, (2) separating immune cells from stromal cells, (3) immune cell subtyping, (4) tumor cell metacluster annotation, and (5) stromal cell metacluster annotation. Cells that did not fit the criteria for any cell types were re-clustered or finally labeled as unknown. Markers used to annotate tumor cells, stromal cells, and immune cell subtypes are presented in Supplementary Fig. 1b.

To confirm the annotations were correct, the identities of the cells were visually confirmed by overlapping the cell class on the original images using Napari image-viewer. The samples were stained for CD3, CD4, and CD8, and scored by a pathologist for the average count of CD3 +, CD4 +, and CD8 + T-cells of three selected CD3 + T-cell rich tumor areas per patient[6], and these annotations were compared with our image-analysis based annotations. Simpson's diversity index (SDI) was calculated using R-package (https://cran.r-project.org/web/packages/vegan/vignettes/diversity-vegan.pdf).

Lineage trajectory analyses for immune cell subtypes and tumor metaclusters were performed using R package STEMNET (https://git.embl.de/velten/STEMNET)[31]. For immune cells, macrophages expressing only either IBA1, CD163, or CD11c were not considered developmental endpoints. For tumor metaclusters, all metaclusters were annotated as developmental endpoints. Markers in the immune cell lineage trajectory plot included immune cell type markers (IBA1, CD163, CD11c, CD4, CD8a, CD3d, FOXP3, CD20) and functional markers (Ki67, P21, PD1, PDL1, cCasp3 and pSTAT1). Markers in tumor metacluster lineage trajectory plot included the functional markers (Ki67, P21, PDL1, cCasp3 and pSTAT1), as well as E-cadherin and vimentin.

The cell-cell spatial neighborhoods were assessed using a cutoff of 30 pixels from the centroid of each cell to the centroid of neighboring cells using rangesearch function in MATLAB, excluding itself from the resulting list. The mean proportion of each cell type neighboring each specific centering cell type was calculated per core, and these values were normalized by dividing the value by the proportion of the specific cell type per core, after which the log2 fold changes were calculated between the *BRCA1/2*mut and HRwt tumors to compare their spatial cell-cell neighborhood frequencies.

**Statistics**. Statistical analyses were performed in R version 3.6.1. Unpaired two-tailed Wilcoxon rank-sum tests were used to calculate two-sided p-values for pairwise comparisons. Effect sizes were calculated for the differences in cell-type proportions between the two mutational groups. False discovery rate was controlled by the Benjamini-Hochberg procedure for functional marker expression differences, for correlation matrix between the proportions of the cell types, and for the spatial analysis of fractions of neighbors for each cell type. Those values passing FDR < 0.1 are discussed in the text. Spearman correlation coefficients and their p-values were used in correlation analyses. An lowess regression was used in situations where linear regression was inappropriate due to violation of assumptions. P-values less than 0.05 were considered significant. Kaplan-Meier graphs were plotted according to the method using the survival and survminer packages in R, and patients at risk with no events at the end of follow-up were

censored. P-values were calculated with the log-rank test. For immune subtype proportions, the median value was used as a cut-off for high- and low proportion groups. For functional marker expressions, the top 1/3 percentile was annotated as high and the rest as low. Cox-proportional hazards were calculated for the first 800 days. The proportional hazards assumption for Cox-proportional hazards were tested using cox.zph function in the survival package in R, and models with values less than 0.05 were considered not suitable for analysis.

**Reporting summary**. Further information on research design is available in the Nature Research Reporting Summary linked to this article.

## Data availability
The datasets generated and analyzed during the current study are publicly available at Synapse ID syn23747228. Source data are provided with this paper. OncoPanel information is under restricted access and may be requested from the authors of Strickland et al.[6] PMID: 26871470. Source data are provided with this paper.

## Code availability
Custom R scripts used to analyze the data are available at Github https://github.com/farkkilab/pubs/tree/master/Launonen_et_al_2021.

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

## Acknowledgements

This study was funded by the Sigrid Jusélius Foundation (A.F.), Finnish cancer society (A.F., J.C.), Academy of Finland (grant number 322979 to A.F.), Paolo Foundation (AF), The Finnish Medical Foundation (A.F., I-M.L), Finnish Cultural Foundation (AF), Instrumentarium Foundation (A.F., J.C.), University of Helsinki (A.F.), NIH grant U54-CA225088 to P.K.S and Ludwig Center at Harvard. Z.M. is supported by NCI grant R50-CA252138. Illustrations for Figs. 1a, 5e, 6 were created with Biorender.com.

## Author contributions

I-M.L analyzed the data and wrote the paper; N.L., J.C., E.A.A., and A.S. analyzed the data and wrote the paper; C.A.J. and J.R.L. performed the experiments and wrote the paper; U-M.H., Z.M., B.E.H., K.C.S., S.S., K.A., A.D.A., P.A.K., and P.K.S. provided materials and resources, supported the data analysis and wrote the paper; A.F. conceived and supervised the study and wrote the paper.

## Competing interests

P.K.S. is a member of the SAB or Board of Directors of Applied Bio-math, Glencoe Software, RareCyte Inc., and has equities in these companies; he is a member of the SAB of NanoString Inc. and a consultant for Merck and Montai Health. P.K.S. has received research funding from Novartis and Merck. None of the above declared relationships has influenced the content of this manuscript. The other authors report no conflicts of interest.
