## [Peer review file · Nature Communications]

Note: This manuscript has been previously reviewed at another journal that is not operating a transparent peer review scheme. This document only contains reviewer comments and rebuttal letters for versions considered at Nature Communications.

REVIEWERS' COMMENTS

Point-by-point response to reviewer comments NATCOMM-21-38938-T

Reviewer's Comments:

Reviewer #1 (Remarks to the Author)

The authors have adequately addressed my concerns.

Reviewer #2 (Remarks to the Author)

The authors have answered my queries well and the manuscript is improved overall.

However the abstract still needs some revision. For instance I am not sure that someone who first reads the abstract will understand the sentence of lines 46-48. Moreover, in the final sentence do they mean 'premise' or 'promise'?

- **We have now modified the abstract lines 46-47 from “Importantly, we found an opposing prognostic role of a proliferative tumor-cell phenotypic subpopulation in the HR-genotypes, which associated with enhanced spatial tumor-immune interactions by the CD8+ and CD4+T-cells in *BRCA1/2*mut tumors.” into “Importantly, we find a prognostic role of a proliferative tumor-cell subpopulation, which associates with enhanced spatial tumor-immune interactions by the CD8+ and CD4+T-cells in *BRCA1/2*mut tumors.” to make it easier for the reader to comprehend.**
- **We have now changed the word premise to potential.**

Reviewer #3 (Remarks to the Author)

I have no more comments.